# Silicon and Zinc Fertilizer Application Improves Grain Quality and Aroma in the *japonica* Rice Variety Nanjing 46

**DOI:** 10.3390/foods13010152

**Published:** 2024-01-02

**Authors:** Xiaodong Wei, Yadong Zhang, Xuemei Song, Ling Zhao, Qingyong Zhao, Tao Chen, Kai Lu, Zhen Zhu, Shengdong Huang, Cailin Wang

**Affiliations:** Institute of Food Crops, Jiangsu Academy of Agricultural Sciences, East China Branch of National Technology Innovation Center for Saline-Alkali Tolerant Rice, Nanjing Branch of China National Center for Rice Improvement, Jiangsu High Quality Rice Research and Development Center, Nanjing 210014, China; weiyinglin@163.com (X.W.); zhangyd@jaas.ac.cn (Y.Z.); songxuemei0414@163.com (X.S.); zhaoling@jaas.ac.cn (L.Z.); qingyong2001@163.com (Q.Z.); chentao19801014@126.com (T.C.); lukai@jaas.ac.cn (K.L.); jsnkyzz@126.com (Z.Z.); 19980010@jaas.ac.cn (S.H.)

**Keywords:** rice, silicon, zinc fertilizer, grain quality, aroma

## Abstract

This study examined how silicon and zinc fertilizers affect the quality and aroma of Nanjing 46. We applied nine different fertilizer treatments, one involving soil topdressing at the top fourth leaf-age stage and one involving foliar spraying during the booting stage of the silicon and zinc fertilizers. We tested the effects of the nine treatments on grain quality and aroma. Silicon and zinc fertilizers significantly affected the brown rice rate, milled rice rate, head rice rate, amylose content, gel consistency, RVA characteristic value, taste value, and aroma but did not affect the chalky grain rate, chalkiness, protein content, rice appearance, hardness, stickiness, balance, peak time, or pasting temperature. Silicon fertilizer decreased the rate of brown rice and milled rice, whereas zinc fertilizer increased the rate of brown rice and milled rice. Silicon and zinc fertilizers improved the head rice rate. Compared to silicon fertilizer, the impact of zinc fertilizer on increasing the head rice rate was more pronounced. Although the effects of silicon and zinc fertilizers on the amylose content and RVA characteristic value varied depending on the treatment, their application could lower the amylose content, increase gel consistency, improve breakdown viscosity, decrease setback viscosity, increase aroma, and improve the taste value of rice.

## 1. Introduction

Rice is the staple food of more than 50% of the world’s population [1]. China is the largest producer of rice worldwide. Since the founding of the People’s Republic of China, rice varieties and their yields have continued to improve [2], ensuring that the supply of rice meets the demands of China’s growing population. Since the beginning of the 21st century, rice varieties have improved, with increased yields and improved quality. Numerous rice varieties that are high-quality, high-yielding, and multi-resistant have been bred and popularized in production [3]. Nanjing 46 is a high-yielding *japonica* rice variety with excellent eating quality. The Japanese high-quality *japonica* rice Kanto 194 and the Jiangsu high-quality and high-yield *japonica* rice Wuxiangjing 14 were used as male and female parents, respectively, in the breeding process. After screening for appearance and eating quality and molecular marker-assisted selection over several generations, the Jiangsu and Shanghai Crop Variety Approval Committees approved the Nanjing 46 variety in 2008 and 2009, respectively. Nanjing 46 combines the fragrance of the female parent, Wuxiangjing 14, with the soft rice characteristics of the male parent, Kanto 194. The cooked rice has a glossy appearance and a soft, smooth, and elastic texture; it does not harden after cooling and has an excellent taste quality. All primary physical and chemical indices met the national secondary high-quality rice standard. Since its approval, Nanjing 46 has won the “Gold Award” and other honorary titles more than 30 times in the National and Jiangsu Provincial high-quality rice evaluation. It was conferred the “best award” in Japan in 2016. It placed first in the *japonica* rice group and received the “Gold Award” for the second time in the national taste quality evaluation of high-quality rice varieties in 2019 [4]. This approach has become the first choice for obtaining premium rice in the Yangtze River Delta.

Variety is the most important factor affecting rice quality. In addition, the production environment, cultivation practices, fertilizer types, amounts and methods, water management, plant diseases, and pests also impact the quality of rice. To ensure the excellent taste quality of Nanjing 46, we developed technical regulations for the quality and cultivation of Nanjing 46 [5] and implemented standardized technical control throughout the entire process, from seeding to seedling raising, transplanting fertilizer and water management, and harvesting. However, years of planting in Nanjing 46 revealed that the same standardized cultivation method and location yielded different taste qualities in subsequent years; specifically, the fragrance gradually weakened over time.

Silicon is an essential element for rice growth. Its importance is only after nitrogen, phosphorus, and potassium [6]. Zinc functions as a nutrient manager in plants, coordinating the distribution of nutrients throughout the plant body. Rice plants are susceptible to zinc deficiency. Zinc deficiency delays rice growth and development, reduces rice stress resistance, and decreases the number of tillers, resulting in lower yields [7]. Topdressing silicon and zinc fertilizers affected rice yield [8,9], rice appearance, processing and eating quality [10,11], flavor content [12,13], diseases and pests, and stress resistance [14,15]. As a result, research on the effects of silicon and zinc fertilizer on rice quality is critical for optimizing high-quality rice varieties and integrating cultivation techniques. Applying silicon and zinc fertilizer at the second leaf-age from the top could significantly improve the eating quality of Nanjing 9108, increase the rate of brown rice, milled rice, and head rice, decrease the chalky grain and chalkiness rate of rice, and improve its processing and appearance quality. At the same time, the gel consistency, peak viscosity, setback viscosity, taste, and taste value of Nanjing 9108 improved [16]. Applying silicon and zinc fertilizer to the soil at a base level may encourage the synthesis of 2-acetyl-1-pyrroline (2-AP) in rice, thus significantly increasing the content of aroma substances in rice grains [17,18,19]. Silicon fertilizer applied via foliar spraying had a significant impact on the amylose content of rice in addition to being able to raise grain protein levels [20]. Zinc fertilizer foliar spraying could also increase the content of aroma substances in fragrant rice [21].

There have been some studies in recent years on the response of characteristics like yield and rice quality of *japonica* rice with good taste to silicon and zinc fertilizers in the middle and lower reaches of the Yangtze River [16]. Soil topdressing is the most studied method of fertilizer application. The research on the effect of foliar spraying is less extensive, and the results of different varieties vary. There was no comparison of the effects of silicon and zinc fertilizers and their application methods on aroma and application effect. Therefore, in this study, Nanjing 46 was used to investigate the effects of silicon and zinc fertilizers and their different application methods on the quality and flavor of rice to provide a reference basis for optimizing and cultivating aroma-enhanced Nanjing 46.

## 2. Materials and Methods

### 2.1. Experimental Material

Nanjing 46, a *japonica* rice variety with good taste, was used as the research material. It is widely cultivated in the middle and lower reaches of the Yangtze River of China, with a growth cycle of 165–170 days.

### 2.2. Experimental Site

The trial was conducted from 2019 to 2020 at Lishui Plant Science Base of Jiangsu Academy of Agricultural Sciences, which has a subtropical monsoon climate. The soil is sandy loam with a medium fertility level. It contains 1.25 g kg^−1^ total nitrogen, 92.3 mg kg^−1^ alkali hydrolysable nitrogen, 35.5 mg kg^−1^ available phosphorus, 86.9 mg kg^−1^ available potassium, 0.85 mg kg^−1^ available zinc, and 110.48 mg kg^−1^ available silicon.

### 2.3. Experimental Treatment

The experiment employed soil topdressing and foliar spraying techniques. The topdressing occurred at the 4th leaf-age stage from top, and the foliar spraying occurred at the booting stage (5~7 days before heading). We used a total of 9 treatment levels (Table 1). 

The silicon fertilizer for topdressing was an instant silicon fertilizer from Lianyungang Fulong Agricultural Development Co., Ltd. (Lianyungang, China), with Si content of 25%, and the dosage was 30 kg hm^−2^. The zinc fertilizer used for topdressing was 98% ZnCl_2_, and the dosage was 15 kg hm^−2^. Silicon and zinc fertilizers were mixed with fine soil at a rate of 150 kg hm^−2^ and evenly distributed. The silicon fertilizer sprayed on the leaf containing Si ≥ 50% was from Chuzhou Geili Fertilizer Technology Co., Ltd. (Chuzhou, China), and the dosage used was 1.5 kg hm^−2^. The zinc fertilizer sprayed on the leaf was 98% ZnCl_2_, and the dosage was 1.8 kg hm^−2^. All treatments were repeated three times, with each replicate containing nine randomly placed plots measuring 16 m^2^. A ridge was created between each plot and covered with a plastic sheet, allowing for separate irrigation and drainage.

### 2.4. Experimental Management

Seeds were sown on May 14 and transplanted on June 10 each year. Seedlings were raised on plastic trays and transplanted by hand, with plant and row spacing of 13 cm × 30 cm and four seedlings per plant. The total amount of nitrogen fertilizer was 240 kg hm^−2^ of pure nitrogen, with a 4:4:2 ratio of base fertilizer, tillering fertilizer, and panicle fertilizer. To prepare the soil, 450 kg hm^−2^ of compound fertilizer with N-P_2_O_5_-K_2_O contents of 20%, 12%, and 16% was used as the base fertilizer. Seven days after transplanting, 120 kg hm^−2^ of urea was applied, followed by 75 kg hm^−2^ of urea seven days later. Panicle fertilizer was applied at the top fourth leaf-age stage using 300 kg hm^−2^ compound fertilizer with an N-P_2_O_5_-K_2_O content of 20%-12%-16%. Once the tillers reached 80% of the expected panicles, the field was drained and roasted. Other cultivation and management practices such as irrigation, disease control, pest control, and weed control followed the recommended guidelines of high-yield cultivation by local government agencies.

### 2.5. Determination Items of Parameters and Methods

At maturity, rice from each plot was harvested and manually threshed. The rice grains were subsequently sun-dried to a moisture level of approximately 14% for the determination.

#### 2.5.1. Determination of Rice Quality

Measurement of the rate of brown rice, milled rice, and head rice: 30 g of rice grains (W_0_) was randomly weighed, husks were removed with an experimental huller, and the weight was measured (W_1_); the rice was ground into polished rice using a rice miller and weighed (W_2_); and after removing the broken rice, the weight was measured again (W_3_). The brown rice percentage, milled rice percentage, and head rice percentage were calculated using the following formulas. The measurements were repeated three times, and the average value was calculated.
Brown rice (%) = W_1_/W_0_ × 100%
Milled rice (%) = W_2_/W_0_ × 100%
Head rice (%) = W_3_/W_0_ × 100%

Measurement of chalky grain and chalkiness: 100 grains (N_0_) were randomly selected from the head rice, all the grains were selected with chalkiness (N_1_), and 10 grains (if the total chalky grains were less than 10 grains, the actual number was taken) were selected from the chalky rice. The chalky rice grains were laid flat, the percentage of chalky area to the entire rice grain area was visually measured, and the average chalky size (W_D_) was calculated for the chalky rice grains. The chalky grain percentage and chalkiness degree were calculated according to the following formulas: The measurements were repeated three times, and the average value was calculated.
Chalky grain (%) = N_1_/N_0_ × 100%
Chalkiness (%) = W_D_ × N_1_/N_0_ × 100%

The gel consistency (GC) of the milled rice was determined according to the national standard of the People’s Republic of China (GB/T 17891–1999): “High-quality paddy”. The amylose content (AC) of the rice flour was determined according to the standard NY/T 83–2017 issued by the Ministry of Agriculture and Rural Affairs. We obtained four reference samples (AC: 1.5%, 10.6%, 16.4%, and 25.6%) from the China Rice Research Institute. The protein content (PC) was determined by Kjeltec 8400 (FOSS) and multiplied by the coefficient of 5.95 [22].

#### 2.5.2. Determination of the RVA Profile Characteristics

We used an RVA viscosity tester (TechMaster, Perten, Stockholm, Sweden) to measure the viscosity of the head rice flour. The heating profile was set up according to the American Association of Grain Chemistry’s operating guidelines (AACC61-01 and 61-02). Briefly, 3 g of rice flour was put into an aluminum can and then mixed with 25 mL of distilled water. Once the RVA cycle started, the samples were stirred at 960 rpm by the plastic paddle, after which the rotation speed was reduced to 160 rpm for the remainder of the experiment. The samples were heated from 50 to 95 °C and then cooled back to 50 °C. The primary parameters automatically measured by the instrument included peak viscosity (PV), hot viscosity (HV), final viscosity (FV), pasting temperature (PaT), and peak time (PeT). The secondary parameters, namely, breakdown viscosity (BDV = PV − HV), setback viscosity (SBV = FV − PV), and recovery viscosity (CSV = FV − HV), were calculated. Every sample was measured three times, and the average result of those three measurements served as the value for that sample.

#### 2.5.3. Determination of the Taste of Cooked Rice

The appearance, hardness, stickiness, balance, and taste value of rice were measured automatically using a rice taste meter (STA-1A, Satake Company, Hiroshima, Japan).

#### 2.5.4. Determination of 2-AP

The 2-acetyl-1-pyrroline (2-AP) in the milled rice was extracted with absolute ethanol and chloroform, and the 2-AP content was determined using a Thermo TSQ 8000 EVO mass spectrometer (Thermo Fisher Scientific Inc., Waltham, MA, USA).

### 2.6. Data Analysis

Statistical analysis was conducted using SPSS Statistics software (version 16.0, Chicago, IL, USA). Two-way year and treatment analysis of variance (ANOVA) was performed to identify significant differences (*p* ≤ 0.05) among the years, treatments, and their interactions. Due to the differences between year divisions and the interactions between years and treatments, except for the significant level of 2-AP content (*p* < 0.01), all other traits were not significant. Therefore, take the average of two years for multiple comparisons. The data were presented as the mean ± standard deviation (SD). For the traits with significant differences according to ANOVA, Duncan’s SSR multiple range method was used to compare the differences among treatment means. The effect of each treatment was evaluated in light of how it differed from that of the control (CK). The effects of the different treatments were compared using the difference in average values between the different treatments.

## 3. Results

The variance analysis revealed that most traits significantly varied between treatments, and the changing trend was constant across all treatments. The difference between years was not significant, except for gel consistency and protein content reaching 5% significance and 2-AP content reaching 1% significance. Except for 2-AP content, which reached a significance level of 1%, the interaction effect between years and treatments was not significant, indicating that the two-year experiment was repeatable.

### 3.1. Effects of Silicon and Zinc Fertilizers on the Processing and Appearance Quality of Nanjing 46

There were significant differences among the treatments for brown rice, milled rice, and head rice rate with a 1% probability of occurrence (Table 2). However, there was no significant difference between the rate of chalky grains and the degree of chalkiness.

Multiple comparative analyses of the brown rice, milled rice, and head rice rates with significant analysis of variance revealed that the brown rice rate was the highest in the Si-L + Zn-L treatment and the lowest in the Si-B + Si-L treatment. There were significant differences between the Si-L + Zn-L treatment and the Si-B + Si-L, Si-B, Si-L, and Zn-B treatments, but there were no significant differences among the other treatments. The milled rice rate was the lowest in the Si-B + Si-L treatment and significantly different from that in the other treatments. The head rice rate was greater in the Si-L + Zn-L, Zn-B + Zn-L, and Si-B + Zn-B treatments and lower in the Si-B + Si-L, Si-B, and CK treatments. The difference between the higher treatments and the lower treatments was significant (Table 3).

The difference in values between each treatment and the control defined the effect on processing quality. Compared to those in the control, the percentages of brown rice and milled rice increased slightly in the Zn-L, Zn-B + Zn-L, and Si-L + Zn-L treatments but decreased in the other treatments (Figure 1). Only the Si-B, Si-L, Zn-B, and Si-B + Si-L treatments resulted in significant decreases. In the Si-B + Si-L treatment, the milled rice rate decreased at a 1% significance level. Except for the Si-B and Si-B + Si-L treatments showing a slight decrease, the head rice rate increased compared to that of the control. The increases in the Zn-L, Zn-B + Zn-L, Si-B + Zn-B, and Si-L + Zn-L treatments were significant at 5% or 1%, respectively.

Table 4 compares the effects of various silicon and zinc fertilizer treatments on Nanjing 46 processing quality. Under treatment, except for head rice, the brown and milled rice rates decreased, but the differences were not statistically significant. On the other hand, the head rice rate increased by 3.3 percentage points, reaching a significance level of 5%. In general, applying silicon fertilizer versus zinc fertilizer significantly reduced the processing quality. However, when the effects of silicon and zinc fertilizer were compared using specific application modes, namely soil topdressing or foliar spraying, the processing quality decreased significantly (*p* < 0.05) by 0.6 percentage points only for the brown rice rate in Si-L versus Zn-L. Applying silicon versus zinc fertilizer concurrently via soil topdressing and foliar spraying resulted in a 1.0%, 2.0%, and 7.0% decrease in the percentages of brown rice, milled rice, and head rice, respectively, with a 5% or 1% significance. Compared to that of foliar spraying, the processing quality of soil topdressing decreased, but not significantly, except for a decrease of 0.5 percentage points in the brown rice rate of Zn-B, which was 5% more significant than that of Zn-L. With the simultaneous application of zinc and silicon fertilizer, brown rice and milled rice rates significantly decreased by 0.5 and 0.7 percentage points, respectively, but the decrease in head rice rate was not significant.

### 3.2. Effects of Silicon and Zinc Fertilizers on the RVA Profile of Nanjing 46

The analysis of variance revealed that the RVA profile characteristics reached a significance level of 1%, except for paste temperature, which had a significance level of 5% (Table 2).

Multiple comparisons of the RVA profile showed that the PV was highest in the Zn-L treatment and lowest in the Si-B + Zn-B and Zn-B treatments. The HV was highest in the Si-L treatment and lowest in the Si-B + Zn-B treatment. The FV was highest in the Si-L, Zn-B + Zn-L, and Zn-L treatments and lowest in the Si-B + Zn-B treatment. BDV was highest in the Si-B + Si-L and Si-B + Zn-B treatments and lowest in the Zn-B treatment. The SBV in the Zn-B + Zn-L treatment was the highest, while the SBV in the Si-B + Zn-B and Si-B + Si-L treatments was the lowest. The Zn-B + Zn-L, CK, and Zn-L treatments had higher CSV, while the Si-B + Zn-B and Zn-B treatments had lower CSV values. The PeT for the Zn-B + Zn-L treatment was the highest, and the Si-B + Si-L, Si-B + Zn-B, and Si-B treatments had the lowest PeT values. For PaT, the Si-B + Zn-B, Si-B + Si-L, Si-L, and Zn-B treatments had higher values, while the Zn-B + Zn-L, Si-B treatments combined with CK had the lowest values. Except for the significant difference between the high and low RVA values, the difference in RVA characteristic values among the other treatments was nonsignificant (Table 3).

The effects of different silicon and zinc fertilizers on the RVA characteristics of Nanjing 46 showed that the PV increased significantly by 5% and 1%, respectively, in the Zn-L and Si-L treatments, compared with that in the control. The HV and FV values significantly increased in the Si-L and Zn-B + Zn-L treatments and decreased in the Si-B + Zn-B treatment, resulting in a significant increase in BDV and a significant decrease in SBV in the Si-B + Si-L and Si-B + Zn-B treatments. Except for the Zn-B + Zn-L treatment, the CSV decreased in the other treatments compared with that in the control, with a 5% significant decrease in the Zn-B and Si-B + Zn-B treatments (Figure 2).

The effects of different silicon and zinc fertilizer treatments on the RVA profile of Nanjing 46 are compared in Table 4. Except for the SBV and CSV, the RVA characteristic values of the silicon zinc fertilizer treatments increased but were not significantly different from those of the control. In comparison to those in the zinc fertilizer treatment, the PV, BDV, HV, FV, SBV, and CSV in the silicon fertilizer treatment did not change significantly. Compared to the Zn-B treatment, the Si-B treatment significantly increased PV and BDV while decreasing the SBV. Compared to the Zn-L treatment, the Si-L treatment decreased the PV, BDV, and CSV while increasing the HV, FV, and SBV. BDV in the Si-B + Si-L treatment increased significantly compared with that in the Zn-B + Zn-L treatment, and the other RVA characteristics decreased at a significance level of 5% or 1%, except for PV. Except for a slight increase in BDV, all the other parameters decreased in response to soil topdressing compared to foliar spraying, but the differences were not statistically significant. Only the PV, BDV, and CSV in the Zn-B treatment were significantly lower than those of the RVA profile in the Zn-L treatment, while the other differences were not significant.

### 3.3. Effects of Silicon and Zinc Fertilizers on the Cooking Quality and Aroma of Nanjing 46

According to the analysis of variance results, the differences between treatments were significant at the 1% level for taste value (TV), amylose content (AC), gel consistency (GC), and 2-AP content but not for rice appearance, hardness, stickiness, balance, or protein content (PC). For the PC, GC, and 2-AP content, the differences between years were significant at the 5% or 1% level; however, the 2-AP content exhibited a significant interaction effect between years and treatments only at the 1% level (Table 2).

We performed multiple comparisons with significant analysis of variance for eating quality (Table 3). Except for Zn-B + Zn-L, which had the highest TV, only the control and the Si-L + Zn-L treatments had significantly different TVs. The Si-B + Si-L treatment had the lowest AC, followed by the Si-L treatment, while the Zn-L treatment had the highest AC. There was no significant difference among the other treatments. The Zn-B treatment had the highest GC, while the control had the lowest value. The Si-L treatment had the second lowest GC, while the other treatments had higher GC than the control. The 2-AP content in the Zn-L treatment was the highest and significantly different from that in the other treatments, and that in the Si-L treatment was the lowest. There was no significant difference among the other treatments except for the Si-B + Si-L treatment.

According to the treatment effects of various silicon and zinc fertilizers on the taste quality of Nanjing 46 (Figure 3), all the treatments had higher TV and GC than the control. Apart from the Si-L + Zn-L treatment, which reached a statistically significant difference of 5%, all the treatments had a significantly higher TV than the control. The Zn-L treatment had a significantly greater AC than the control, and the Zn-B + Zn-L treatment had a marginally greater AC. While the AC levels in the Si-L + Zn-L treatment were nearly identical to those in the CK treatment, the AC levels in the other treatments decreased, with the Si-B + Si-L and Si-L treatments reaching significant levels of 1%. The 2-AP content in all the other treatments was greater than that in the control, except for that in the Si-L treatment, which was slightly lower. All the increases were significant, with the relative content of 2-AP in the Zn-L treatment increasing to more than 50%.

By comparing the results of the various silicon zinc fertilizer treatments (Table 4), we determined that TV increased by 3.0 points, GC increased by 7.6 mm, 2-AP content increased by 0.0561 µg/g^−1^, and AC decreased slightly but not significantly. These increases were significant at the 1% level. Similarly, compared with those in the zinc fertilizer treatment, the TV and AC in the silicon fertilizer treatment decreased slightly, while the GC and 2-AP content significantly decreased at 5% and 1% levels, respectively. There was no significant difference between the Si-B and Zn-B treatments. At 1%, the AC, GC, and 2-AP content in the Si-L treatment were significantly lower than those in the Zn-L treatment. The Si-B + Si-L treatment had lower TV, AC, GC, and 2-AP content than the Zn-B + Zn-L treatment, and AC and 2-AP content reached a 1% significance level. Compared with those under foliar spraying, the TV and AC decreased slightly with soil topdressing, while the GC and 2-AP content increased slightly. However, these differences were not statistically significant. The Si-B treatment had 5% and 1% greater AC and 2-AP content, respectively, than the Si-L treatment. The AC and 2-AP content were significantly lower in the Zn-B treatment than in the Zn-L treatment.

Silicon zinc fertilizer combinations could significantly increase the flavor (2-AP), increase the GC, and improve the TV. Zinc fertilizer application had a more significant effect on flavor than silicon fertilizer. Soil topdressing with silicon fertilizer significantly outperformed foliar spraying. When zinc fertilizer was applied, foliar spraying performed significantly better than soil topdressing. When applied as a soil topdressing, silicon fertilizer had effects that were noticeably superior to those of zinc fertilizer, and the effects of zinc fertilizer as a foliar spray were noticeably better than those of silicon fertilizer. With respect to the combined application of soil topdressing and foliar spraying, zinc fertilizer yielded better results than silicon fertilizer.

## 4. Discussion

The effects of silicon and zinc fertilizers on the rate of brown rice, milled rice, and head rice, the rate and degree of chalky grain, as well as the processing and appearance quality, have been the subjects of several studies. Silicon fertilizer could significantly increase the percentage of brown, milled, and head rice when the application rate was 0–300 kg/hm^2^ while decreasing the chalkiness and chalky grain rate [23,24]. Zinc application could significantly increase the head rice rate of Guixiangzhan and reduce its chalky grain rate [25]. A different study showed that silicon fertilizer applied at concentrations between 60 and 120 kg/hm^2^ causes a reduction in the quality of the processing and appearance [26]. The application of zinc and silicon fertilizers during the transplanting, effective tillering, and the top fourth and second leaf-age stages can lower the rate of brown rice, increase milled rice and head rice rates, and decrease the rate and degree of chalkiness [16]. According to Zhao et al. [27], silicon treatment improved processing quality while degrading appearance quality and considerably increasing the brown rice rate, milled rice rate, head rice rate, chalky grain rate, and chalkiness. Silicon fertilizers can enhance the eating quality and nutritional value; however, observations of these fertilizers vary depending on the time of application. Studies have shown that applying silicon fertilizer at the booting stage can increase the amylose and protein contents of rice grain [27,28]. However, another group demonstrated that the appropriate application of silicon fertilizer could increase the protein content of rice, but the amylose content did not significantly change [24]. Zhang et al. [23] reported that silicon fertilizer can reduce the amylose content while increasing the protein content in rice. The eating quality of rice is closely related to the characteristic value of the RVA profile. The appropriate application of silicon fertilizer in rice can improve peak viscosity, gel consistency, and breakdown viscosity, reduce gelatinization temperature and setback viscosity, improve eating value, and improve cooking and eating quality. Although the effects of zinc and silicon fertilizers on protein and amylose contents vary depending on treatment and time, zinc and silicon fertilizers can still improve the gel consistency, taste, and taste value of rice, as well as its peak and breakdown viscosities and lower setback viscosity [16].

Our results support earlier findings that zinc fertilizer tends to increase the rate of brown rice and milled rice, while silicon fertilizer decreases these rates [16]. In rice, silicon is mainly distributed in the leaves, stems, and glumes, with the glumes having the highest concentration. It is possible that the silicon application-related decreases in brown rice and milled rice rate and the increase in 1000-grain weight are caused by the increase in glume silicon accumulation. Zinc fertilizer, on the other hand, enhances the rate of brown rice rate and milled rice, most likely because it increases the nitrogen content, which increases the content of RNA, ribosomes, and RNA in cells and promotes protein synthesis, making grains full and hard and boosting the rate of whole milled rice. Compared to that of silicon fertilizer, the impact of zinc fertilizer on the milled rice rate was more pronounced. In general, silicon zinc fertilizer did not significantly affect rice protein content, appearance, hardness, viscosity, balance, peak time, or paste temperature but did have varying effects on the values of AC and RVA characteristic parameters. The use of silicon zinc fertilizers also significantly increased GC and taste value.

The results of this experiment indicate that, in terms of enhancing processing and eating quality, the effects of a single application of silicon and zinc are comparable, the effect of a foliar spray is marginally superior to that of a soil topdressing, and the combined effect of a soil topdressing and foliar spray, and that of silicon and zinc is comparable to that of a single application. However, the impact of zinc fertilizer on the processing and eating quality was significantly greater. For silicon fertilizer, soil topdressing had a much greater impact than foliar spraying, whereas, for zinc fertilizer, the opposite was true. The effects of silicon fertilizer were noticeably superior to those of zinc fertilizer when applied as soil topdressing, and the effects of zinc fertilizer as a foliar spray were significantly greater than those of silicon fertilizer. For the combined application of soil topdressing and foliar spraying, zinc fertilizer performed noticeably better than silicon fertilizer.

## 5. Conclusions

The application of silicon zinc fertilizer may result in a reduction in the rate of brown and milled rice, an increase in the rate of head rice, a decrease in the amylose content, an increase in gel consistency and breakdown viscosity, a decrease in setback viscosity, an increase in 2-AP content, and an increase in taste value. Zinc fertilizer performed substantially better than silicon fertilizer. Compared with foliar spraying, soil topdressing had a significantly better effect on silicon fertilizer use. When zinc fertilizer was applied, foliar spraying had a much greater effect than soil topdressing.

## Figures and Tables

**Figure 1 foods-13-00152-f001:**
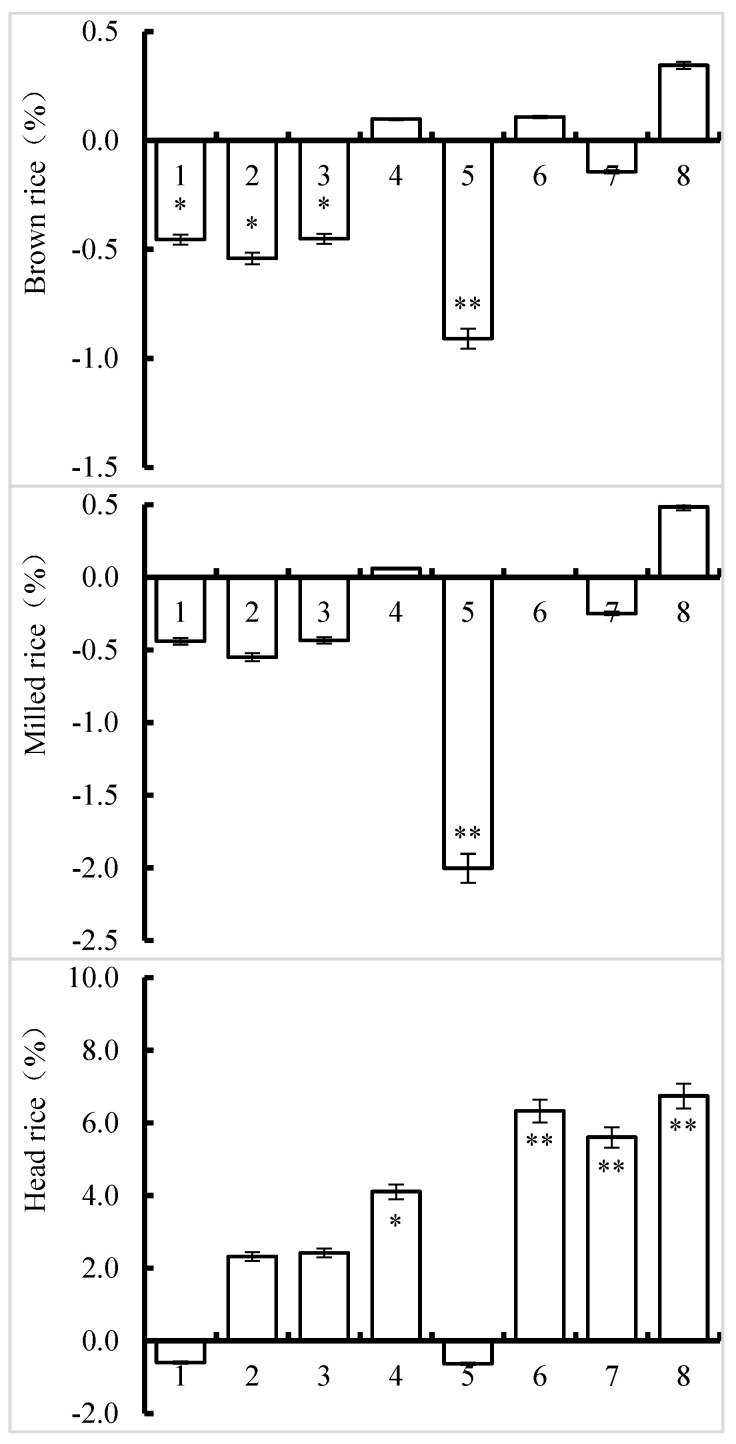
Effect of silicon and zinc fertilizer combinations on processing quality of Nanjing 46. The data in the figure are the average differences between treatments and CK. * and ** represent significant differences at 5% and 1%, respectively. One to eight indicate the Si-B, Si-L, Zn-B, Zn-L, Si-B + Si-L, Zn-B + Zn-L, Si-B + Zn-B, and Si-L + Zn-L treatments, respectively, compared to the control.

**Figure 2 foods-13-00152-f002:**
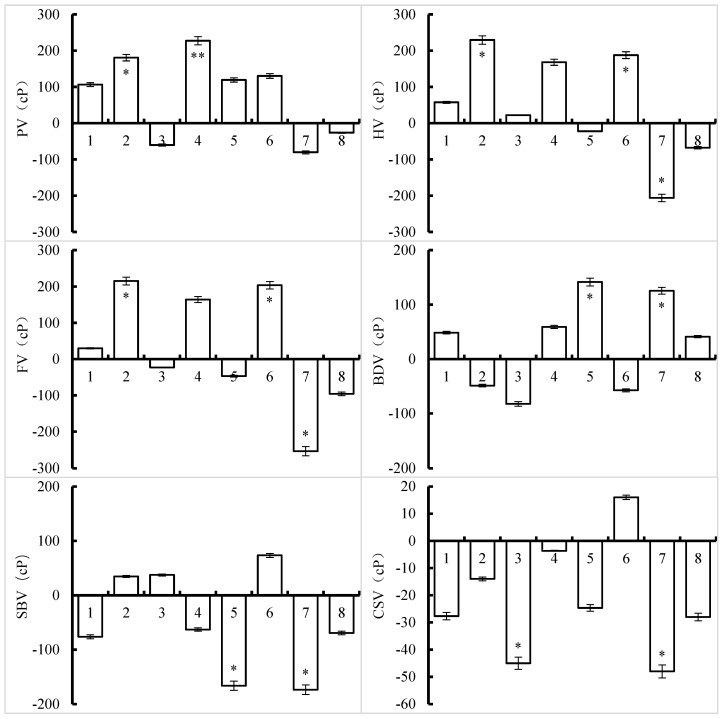
Effects of silicon and zinc fertilizer combinations on the RVA profile of Nanjing 46. The data in the figure are the average differences between treatments and CK. * and ** represent significant differences at 5% and 1%, respectively. One to eight indicate the Si-B, Si-L, Zn-B, Zn-L, Si-B + Si-L, Zn-B + Zn-L, Si-B + Zn-B, and Si-L + Zn-L treatments, respectively, compared to the control.

**Figure 3 foods-13-00152-f003:**
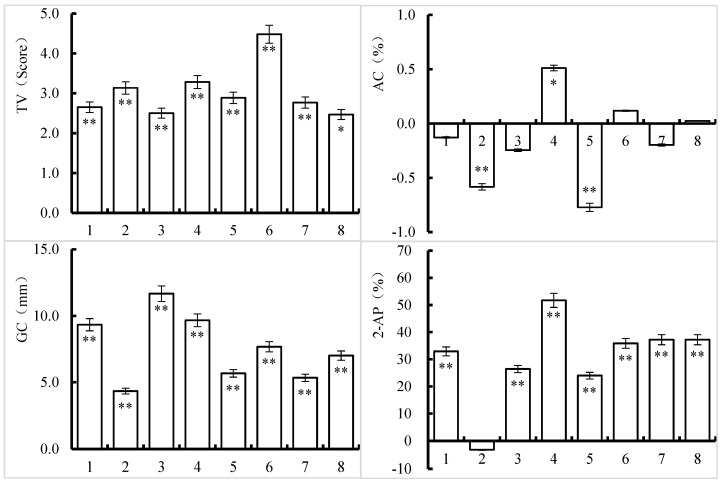
Effects of silicon and zinc fertilizers on eating quality and 2-AP content in Nanjing 46. The data in the figure are the average difference between treatments and CK, in which the content of 2-AP is expressed as a percentage difference. * and ** represent significant differences of 5% and 1%, respectively. One to eight indicate the Si-B, Si-L, Zn-B, Zn-L, Si-B + Si-L, Zn-B + Zn-L, Si-B + Zn-B, and Si-L + Zn-L treatments, respectively, compared to the control.

**Table 1 foods-13-00152-t001:** Treatments used in the experiment.

Symbol	Treatment
CK	no silicon and zinc fertilizer
Si-B	soil topdressing silicon fertilizer
Si-L	foliar spraying silicon fertilizer
Zn-B	soil topdressing zinc fertilizer
Zn-L	foliar spraying zinc fertilizer
Si-B + Si-L	soil topdressing silicon fertilizer and foliar spraying silicon fertilizer
Zn-B + Zn-L	soil topdressing zinc fertilizer and foliar spraying zinc fertilizer
Si-B + Zn-B	soil topdressing silicon fertilizer and soil topdressing zinc fertilizer
Si-L + Zn-L	foliar spraying silicon fertilizer and foliar spraying zinc fertilizer

**Table 2 foods-13-00152-t002:** Analysis of variance for grain quality of Nanjing 46.

**Source of Variance**	**df**	**Brown Rice (%)**		**Milled Rice (%)**		**Head Rice (%)**	**Chalky Grain (%)**			**Chalkiness (%)**
Repetition	2	0.07		0.53		1.70	0.07			0.12
Year	1	0.11		1.98		0.02	0.26			0.29
Treatment	8	6.03 **		12.90 **		9.73 **	1.10			1.35
Year × treatment	8	0.53		2.58		1.48	1.13			1.21
Source of variance	df	PV (cP)	HV (cP)	FV (cP)	BDV (cP)	SBV (cP)	CSV (cP)		PeT (min)	PaT (°C)
Repetition	2	0.35	0.27	0.28	0.21	0.38	0.77		0.33	1.42
Year	1	0.23	0.96	0.62	0.83	0.51	0.02		0.19	0.08
Treatment	8	6.51 **	6.89 **	4.10 **	7.64 **	4.52 **	6.66 **		7.21 **	3.00 *
Year × treatment	8	1.65	1.95	1.41	2.20	1.46	2.11		2.23	0.87
Source of variance	df	Appearance	Hardness	Stickiness	Balance	TV (Score)	PC (%)	AC (%)	GC (mm)	2-AP (µg/g^−1^)
Repetition	2	2.99	1.15	2.10	2.87	0.21	3.91 *	3.41	2.31	4.26 *
Year	1	0.46	0.38	0.15	0.53	0.01	6.51 *	1.37	7.13 *	76.68 **
Treatment	8	2.14	2.97	1.86	2.29	6.83 **	1.30	7.77 **	14.15 **	1292.03 **
Year × treatment	8	0.91	0.90	0.87	0.98	1.73	0.29	1.11	0.98	18.00 **

The data for each parameter in the table are F values. * and ** represent significant differences at the 5% and 1% levels, respectively.

**Table 3 foods-13-00152-t003:** Rice quality in Nanjing 46 after treatment with silicon and zinc fertilizers.

**Treatment**	**Brown Rice (%)**	**Milled Rice (%)**	**Head Rice (%)**	**PV (cP)**	**HV (cP)**	**BDV (cP)**	**FV (cP)**	**SBV (cP)**
CK	84.3 ± 0.1 ab	75.9 ± 0.2 a	59.5 ± 0.0 b	2588 ± 3 bc	1485 ± 62 abc	1103 ± 65 ab	2095 ± 87 ab	−493 ± 89 ab
Si-B	83.8 ± 0.3 bc	75.5 ± 0.4 a	58.9 ± 0.3 b	2694 ± 78 abc	1542 ± 41 abc	1152 ± 119 ab	2125 ± 32 ab	−570 ± 109 ab
Si-L	83.7 ± 0.1 bc	75.4 ± 0.0 a	61.8 ± 0.0 ab	2769 ± 29 ab	1714 ± 10 a	1055 ± 39 ab	2310 ± 13 a	−459 ± 41 ab
Zn-B	83.8 ± 0.2 bc	75.5 ± 0.3 a	61.9 ± 2.4 ab	2528 ± 109 c	1507 ± 157 abc	1021 ± 47 b	2072 ± 164 ab	−456 ± 54 ab
Zn-L	84.4 ± 0.0 ab	76.0 ± 0.2 a	63.6 ± 3.2 ab	2815 ± 5 a	1653 ± 23 ab	1163 ± 18 ab	2259 ± 23 a	−556 ± 17 ab
Si-B + Si-L	83.4 ± 0.0 c	73.9 ± 1.2 b	58.9 ± 1.8 b	2707 ± 42 abc	1462 ± 130 abc	1245 ± 88 a	2048 ± 136 ab	−660 ± 94 b
Zn-B + Zn-L	84.4 ± 0.2 ab	75.9 ± 0.0 a	65.8 ± 0.9 a	2718 ± 11 abc	1672 ± 34 ab	1046 ± 23 ab	2298 ± 39 a	−420 ± 28 a
Si-B + Zn-B	84.1 ± 0.1 ab	75.7 ± 0.1 a	65.1 ± 0.5 a	2508 ± 168 c	1279 ± 172 c	1229 ± 4 a	1841 ± 210 b	−667 ± 42 b
Si-L + Zn-L	84.6 ± 0.2 a	76.4 ± 0.1 a	66.2 ± 0.6 a	2562 ± 39 bc	1417 ± 114 bc	1145 ± 75 ab	1999 ± 115 ab	−563 ± 76 ab
Treatment	CSV (cP)	PeT (min)	PaT (°C)	TV (Score)	AC (%)	GC (mm)	2-AP (µg/g^−1^)	
CK	610 ± 24 a	6.2 ± 0.1 ab	70.4 ± 0.6 b	86.1 ± 0.1 c	10.4 ± 0.2 abc	84.3 ± 0.5 c	0.1855 ± 0.0048 d	
Si-B	582 ± 10 ab	6.1 ± 0.1 b	70.4 ± 0.5 b	88.8 ± 0.3 ab	10.3 ± 0.0 abcd	93.7 ± 2.6 ab	0.2465 ± 0.0027 bc	
Si-L	596 ± 3 ab	6.3 ± 0.1 ab	71.3 ± 0.0 a	89.3 ± 1.3 ab	9.8 ± 0.0 cd	88.7 ± 2.6 c	0.1795 ± 0.0016 d	
Zn-B	565 ± 7 b	6.2 ± 0.1 ab	71.2 ± 0.1 a	88.6 ± 0.2 ab	10.2 ± 0.0 bcd	96.0 ± 0.7 a	0.2345 ± 0.0080 bc	
Zn-L	606 ± 0 a	6.2 ± 0.0 ab	70.7 ± 0.1 ab	89.4 ± 0.4 ab	10.9 ± 0.2 a	94.0 ± 0.0 ab	0.2815 ± 0.0090 a	
Si-B + Si-L	585 ± 6 ab	6.0 ± 0.2 b	71.3 ± 0.6 a	89.0 ± 0.7 ab	9.6 ± 0.5 d	90.0 ± 1.4 bc	0.2300 ± 0.0085 c	
Zn-B + Zn-L	626 ± 5 a	6.4 ± 0.0 a	70.2 ± 0.2 b	90.6 ± 0.0 a	10.5 ± 0.2 ab	92.0 ± 1.4 abc	0.2520 ± 0.0021 b	
Si-B + Zn-B	562 ± 38 b	6.0 ± 0.1 b	71.5 ± 0.2 a	88.9 ± 1.6 ab	10.2 ± 0.2 bcd	89.7 ± 1.2 bc	0.2545 ± 0.0080 b	
Si-L + Zn-L	582 ± 1 ab	6.2 ± 0.0 ab	71.0 ± 0.2 ab	88.6 ± 0.6 b	10.4 ± 0.2 abc	91.3 ± 2.4 bc	0.2545 ± 0.0080 b	

Different lowercase letters after the same column of data indicate a significant difference of 5% level.

**Table 4 foods-13-00152-t004:** Comparison of the effects of silicon and zinc fertilizer treatments on the grain quality of Nanjing 46.

**Comparison between Treatments**	**Brown Rice (%)**	**Milled Rice (%)**	**Head Rice (%)**	**PV (cP)**	**HV (cP)**	**FV (cP)**	**BDV (cP)**
Treatment and nontreatment ^(1)^	−0.2	−0.4	3.3 *	75	46	24	29
Application of Si and Zn fertilizer ^(2)^	−0.6 *	−0.9 *	−3.9 *	36	−38	−49	74
Si-B and Zn-B	0.0	0.0	−3.0	167 *	36	53	131 *
Si-L and Zn-L	−0.6 **	−0.6	−1.8	−47	61	51	−108
Si-B + Si-L and Zn-B + Zn-L	−1.0 *	−2.0 **	−7.0 **	−11	−210 *	−251 *	199 **
Soil topdressing and foliar spraying ^(3)^	−0.3	−0.4	−1.9	−139	−152	−177	13
Si-B and Si-L	0.1	0.1	−2.9	−74	−172	−185	97
Zn-B and Zn-L	−0.5 *	−0.5	−1.7	−288 **	−146	−187	−142 *
Si-B + Zn-B and Si-L + Zn-L	−0.5 *	−0.7 *	−1.1	−54	−138	−158	84
Comparison between Treatments	SBV (cP)	CSV (cP)	TV (Score)	AC (%)	GC (mm)		2-AP (µg/g^−1^)
Treatment and nontreatment ^(1)^	−50	−22	3.0 **	−0.2	7.6 **		0.0561 **
Application of Si and Zn fertilizer ^(2)^	−85	−11	−0.5	−0.6	−3.2 *		−0.0373 **
Si-B and Zn-B	−114	17	0.1	0.1	−2.3		0.0120
Si-L and Zn-L	98	−10	−0.1	−1.1 **	−5.3 **		−0.1020 **
Si-B + Si-L and Zn-B + Zn-L	−240 **	−41 *	−1.6	−0.9 **	−2.0		−0.0220 **
Soil topdressing and foliar spraying ^(3)^	−38	−25	−0.3	−0.2	1.8		0.0067
Si-B and Si-L	−111	−14	−0.5	0.5 *	5.0		0.0670 **
Zn-B and Zn-L	100	−41 *	−0.8	−0.8 **	2.0		−0.0470 **
Si-B + Zn-B and Si-L + Zn-L	−104	−20	0.3	−0.2	−1.7		0.0000

^(1)^ Difference between the average values of all the treatments except for the control and the average value of the control; ^(2)^ difference between the average values of silicon fertilizer treatments (Si-B, Si-L, and Si-B + Si-L) and the zinc fertilizer treatments (Zn-B, Zn-L, and Zn-B + Zn-L); ^(3)^ difference between the average values of the soil topdressing treatments (Si-B, Zn-B, and Si-B + Zn-B) and the foliar spraying treatments (Si-L, Zn-L, and Si-L + Zn-L).* and ** represent significant differences at 5% and 1%, respectively.

## Data Availability

All data generated or analyzed during this study are included in this manuscript.

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
