# Peer review of "Silicon and Zinc Fertilizer Application Improves Grain Quality and Aroma in the *japonica* Rice Variety Nanjing 46"

_foods, 2024, doi:10.3390/foods13010152_

Round 1

Reviewer 1 Report

Comments and Suggestions for Authors

It is my pleasure to have a reading of the interesting MS focusing grain quality and aroma with the application of B and Zn both soil and folder application addressing the title choose for the MS. Nevertheless, this was a surprise to see the results and also abstracted that................(L 11 to L17) that most of the characters were not significant affected like shakiness, chai grains and protein content, etc. but did affect the milled rice rate, brown rice rate, head rice rate, amylose content, etc., .................... it was very confusing to understand that both quality and aroma are the grain (white) characters or the brown (paddy) character that affected by both B and Zn application if different ways. In this case I would suggest to clear that the effect of the B and Zn on white grain or brown is important and how it contributes in quality and aroma of the desired milled rice used as food which also addressed in the title of the MS. 

Comments on the Quality of English Language

NA

Reviewer 2 Report

Comments and Suggestions for Authors

Please see the attached for more detailed comments.

This is primarily a farming project reporting on the results of variations in farming practice on quality parameters, so it is difficult for me to review some aspects.

I recommend dividing up the ANOVA table and means tables and put them with the relevant topic.

The methods need more detail. Even if not a fully detailed procedure, enough that I can get a general understanding of the procedure used.

This would have been better suited for submission to a farming journal.

Comments on the Quality of English Language

Generally okay, but some bits are hard to follow. 

Go through and sort out some of the presentation things - it should be 4th not 4th, 2nd not 2nd. (Hopefully the superscripting will be still present when you receive these comments.)

Reviewer 3 Report

Comments and Suggestions for Authors

Review of manuscript „Silicon and Zinc Fertilizer Improve Grain Quality and Aroma of Good Taste japonica Rice Variety Nanjing 46”:

The manuscript was written in an interesting way. The research problem raised is an important issue in the context of the growing number of people and the increasing problem of feeding them.

Authors used Nanjing 46 to investigate the effects of silicon and zinc fertilizers and their different application methods on the quality and flavor of rice to provide a reference basis for optimizing and cultivating aroma-enhanced Nanjing 46.

The "Data analysis" subsection is written very poorly and does not exhaustively describe the methods used in the manuscript. Moreover, and most importantly, it does not provide the possibility of repeating the experience. This must be described in detail.

Table 1. The full analysis of variance is not presented here (as is clear from the title of the table). Please complete or specify the values given in the table.

Table 2: It should be supplemented with standard deviations and the method used to determine homogeneous groups should be provided.

Table 3: What statistical test was used here?

Figures 1-3: Must be supplemented with a bars.

The authors did not attempt to analyze correlations of features or take a multidimensional look at treatment dependencies. Such analyzes would enrich the manuscript by giving the results a more practical aspect.

Comments:
Citations are not provided as required by the journal.

Paper needs major revision.

Round 2

Reviewer 2 Report

Comments and Suggestions for Authors

I thank the authors for their careful and detailed edits. A few minor things, and it is ready (at least from my perspective).

Comments on the Quality of English Language

The English is fine. One or two little points to clarify.

Author Response

Response to Reviewer 3

Thank you for reviewing the manuscript again. We have made revisions to the manuscript based on your suggestion. The highlighted parts in the original manuscript are the areas of this revision. Please kindly review them again. Thank you!

Line 41-42 :

Original:The cooked rice is shiny with oil, soft and smooth in taste, elastic, does not harden after cooling, and has an excellent taste quality.

Revised:The cooked rice has a gloss appearance, has a soft, smooth and elastic texture, does not harden after cooking, and has an excellent taste quality.

Line 82-83 : 

Original:It is only after in importance to nitrogen, phosphorus, and potassium in the growth and development of rice [6].

Revised:It is only after to nitrogen, phosphorus, and potassium in importance in the growth and development of rice [6].

Line 87 : 

Original:Topdressing

Revised:Top dressing

Line 95 : 

Your question:I don't understand how this variety fits into the discussion. The rest is on Nanjing 46.

Answer:Because "Applying silicon and zinc fertilizer at the 2nd leaf-age ……and taste value of Nanjing 9108 improved", this paragraph describes the research results of Wang et al. (reference [16]), who used Nanjing 9108 as their experimental material. The original manuscript may have inaccurate literature annotations, and the literature annotations have been moved to the end of this paragraph. And, Nanjing 9108, like Nanjing 46, is an excellent taste variety with low amylose content widely planted in the Yangtze River Basin. So, we believe that the research results of Nanjing 9108 can also be used as a reference.

Line 114 : 

Original:It was widely cultivated

Revised:It is widely cultivated

Line 171 : 

Original:four seedlings per hill.

Revised:four seedlings per plant.

Line 219 : 

Original:Thirty grams of rice

Revised: 30 g of rice

Line 250: 

Original:aluminum tank,

Revised:aluminum can,

Line 251-252 : 

Original:Initially, the samples were stirred in the instrument cell using a plastic paddle 251 rotating at 960 rpm, after which

Revised:Once the RVA cycle started, the samples were stirred at 960 rpm by the plastic paddle, after which

Table 3

Your question:Normally these are superscript

Answer:Yes, there are also superscripts marked, but I haven't seen MDPI papers marked with superscripts.

And, Table 3 should be annotated with: Different lowercase letters after the same column of data indicate the significant difference of 5% level.

Line 702: Abbreviations 

Original:Rapid visco-analyzer

Revised:Rapid Visco-analyzer

Reviewer 3 Report

Comments and Suggestions for Authors

The Authors have significantly improved the manuscript. They took into account my comments and suggestions. The manuscript can be published in the current version.

Author Response

Thank you for reviewing the manuscript again. We have made minor revisions to the English version of the manuscript based on expert feedback. The highlighted parts in the original manuscript are the areas of this revision. Please kindly review them again. Thank you!
